# The Potentiation of Anti-Tumor Immunity by Tumor Abolition with Alpha Particles, Protons, or Carbon Ion Radiation and Its Enforcement by Combination with Immunoadjuvants or Inhibitors of Immune Suppressor Cells and Checkpoint Molecules

**DOI:** 10.3390/cells10020228

**Published:** 2021-01-25

**Authors:** Yona Keisari, Itzhak Kelson

**Affiliations:** 1Department of Clinical Microbiology and Immunology, Sackler Faculty of Medicine, Tel Aviv University, Tel Aviv 6997801, Israel; 2Sackler Faculty of Exact Sciences, School of Physics and Astronomy, Tel Aviv University, Tel Aviv 6997801, Israel; kelson@tauex.tau.ac.il

**Keywords:** alpha radiation, proton radiation, carbon ion, immunotherapy, immunoadjuvants, checkpoint inhibitors, immune suppressor cells, toll-like receptors, tumor abolition, anti-tumor immunity

## Abstract

The delivery of radiation therapy (RT) for cancer with intent to cure has been optimized and standardized over the last 80 years. Both preclinical and clinical work emphasized the observation that radiation destroys the tumor and exposes its components to the immune response in a mode that facilitates the induction of anti-tumor immunity or reinforces such a response. External beam photon radiation is the most prevalent in situ abolition treatment, and its use exposed the “abscopal effect”. Particle radiotherapy (PRT), which has been in various stages of research and development for 70 years, is today available for the treatment of patients in the form of alpha particles, proton, or carbon ion radiotherapy. Charged particle radiotherapy is based on the acceleration of charged species, such as protons or carbon-12, which deposit their energy in the treated tumor and have a higher relative biological effectiveness compared with photon radiation. In this review, we will bring evidence that alpha particles, proton, or carbon ion radiation can destroy tumors and activate specific anti-tumor immune responses. Radiation may also directly affect the distribution and function of immune cells such as T cells, regulatory T cells, and mononuclear phagocytes. Tumor abolition by radiation can trigger an immune response against the tumor. However, abolition alone rarely induces effective anti-tumor immunity resulting in systemic tumor rejection. Immunotherapy can complement abolition to reinforce the anti-tumor immunity to better eradicate residual local and metastatic tumor cells. Various methods and agents such as immunoadjuvants, suppressor cell inhibitors, or checkpoint inhibitors were used to manipulate the immune response in combination with radiation. This review deals with the manifestations of particle-mediated radiotherapy and its correlation with immunotherapy of cancer.

## 1. Introduction

Cancer is the second leading cause of death globally, affecting 19 million new patients every year. About one in six deaths is due to cancer, and it was responsible for an estimated 9.6 million deaths in 2018 [1].

Patients are treated primarily by surgery, chemotherapy, biological (immunotherapy, angiogenesis inhibitors, cancer growth inhibitors), and radiation therapy. Surgery and radiotherapy are mainly used for debulking the primary tumor, while chemotherapy and immunotherapy are used for both local tumor control and antimetastatic treatment.

Systemic anti-tumor treatments are aimed primarily to eliminate distant metastases, which are responsible for the death of at least 67% and up to 90% of cancer patients, depending on the tumor type [2].

The main systemic antimetastatic treatment, chemotherapy, is not specific and affects mostly replicating cells, while resting cancer cells may not be destroyed. To better control metastatic cancer, the host immune system should be stimulated. Yet, successful specific stimulation of the immune system against tumors by active immunization was seldom achieved, even in antigenic tumors. As an alternative, attempts were made to engage the host immune system in situ to react against tumor-associated antigens by making the tumor its own vaccine, without the need to isolate tumor-0associated antigens [3]. This approach has its roots in a phenomenon first described as the “abscopal effect” [4,5]. It has been demonstrated that aggressive in situ tumor destruction (abolition) could lead to the release of tumor antigens, which can stimulate anti-tumor immune responses, which will destroy residual malignant cells in primary tumors and distant metastases [6].

## 2. Tumor Immunology and Immunotherapy of Cancer

Immunotherapy of cancer has been a major goal since 1891 when William Coley started an experimental treatment of cancer patients with bacterial-derived products, actually introducing the first danger signal treatment, claiming that the beneficial effect is a result of activation of anti-tumor immunity [7]. This effort has constantly grown in the last 130 years, and the number of immuno-oncology drugs under investigation in 2020 had increased by 233% with respect to those in 2017 [8].

The role of the immune response in tumor development and treatment is a very complicated issue. There are many elements involved, such as the variety of tumor types, the organism genetics, the complexity of the immune response, metabolism, age, and the microbiome, to name a few. The magnitude of this field can be understood by the number of publications listed to date (Pubmed search of “Tumor immunology” will yield over 300,000 papers with over 45,000 reviews; “Immunotherapy” 95,000 articles, and 29,000 reviews). A recent comprehensive review has provided a good summary of the field [9].

## 3. Charged Particle Beam Radiotherapy

Along with surgery and chemotherapy, radiation therapy is one of the most important methods of cancer treatment, and approximately 50–70% of cancer patients will receive radiation therapy for curative or palliative intent. The delivery of radiation therapy (RT) for cancer, dating back to the discovery of X-rays in 1895, with intent to cure has been optimized and standardized over the last 80 years.

Radiation therapy involves photons (e.g., gamma-rays and X-rays) or particles (e.g., protons, neutrons, alpha particles, heavy ions, and electrons). The most prevalent radiation treatment is external beam radiation therapy (EBRT) utilizing gamma- or x-rays radiation. Radiation is useful for treatment of local and regional disease sites, or where surgical excision of the tumor is not feasible due to the size and site of tumor, or the patient’s medical condition. The effectiveness of photon radiation is limited mainly due to hypoxia in the tumor.

Radiation induces several cell damage responses, including apoptotic cell death, mitotic catastrophe, necroptosis, ferroptosis, and senescence [10], some of them initiated by DNA double strand breaks (DSBs), which are known to be the most toxic and threatening of the various types of breaks that may occur to the DNA [11].

### 3.1. Proton and Carbon Ion Radiotherapy

Particle radiotherapy (PRT) is an alternative approach that has been practiced in various stages of research and development for 70 years. Today, clinical treatment is available in the form of either proton or carbon ion radiotherapy. Charged particle radiotherapy is based on the acceleration of charged species, such as protons or carbon-12, which deposit their energy in the treated tumor. The particular feature that renders this modality practical is the dependence of the energy loss rate on the particle energy. As the charged particle slows down in matter, the energy loss per unit length increases sharply and a large energy density peak (the Bragg peak) is formed at the end of the trajectory [12]. The deposited energy density at the Bragg peak is typically higher by a factor of 3–5 than it is along the rest of the charged particle path. In addition, the high relative biological effectiveness (RBE) of charged particles relative to photons or electrons makes this modality inherently useful.

The coverage of a well-defined volume of tumor mass is attained by a simple manipulation of the charged particle beam. The volume is essentially divided into voxels, which are three-dimensional cubes with defined coordinates. The control of the lateral position in the target area is done by electromagnetic steering of the beam. The depth of penetration is achieved by varying the beam energy, so that the Bragg peak occurs at the desired voxel. The use of passive shielding as a means to better conform to the treated volume is still used to some extent. In order to minimize the undesired effect on healthy tissue, the irradiation is performed from different directions, much like the customary application of external photon therapy; the toxic effect is distributed over a large volume of healthy tissue, while the therapeutic dose is concentrated in the target volume. As with external photon therapy, it is customary to divide the therapy into a number of fractions, to allow the irradiated healthy tissue to recover from the unavoidable effect of the radiation. In fact, healthy tissue is actually better preserved with charged particle radiotherapy than with photon therapy.

The ideal candidate for PRT is a well-defined, compact tumor situated such that it can be kept motionless during the irradiation and its environment does not change between treatment sessions. Brain tumors were among the first to be treated for these very reasons, and have had very good therapeutic outcomes [13]. Consequently, the number of facilities has grown considerably over the years, with about 100 proton facilities and about 10 carbon facilities around the world.

PRT has some inherent challenges that need to be met. Treatment of nonstationary tumors, such as lung tumors, require the ability to modify the control of the beam to follow the corresponding organ movement. Variations in the body environment between fractions require elaborate beam energy re-calibration to avoid undesirable displacement of the Bragg peak. Reactions along the beam path may result in harmful secondary radiation, whose effect must be considered. Finally, one should note that the number of patients that can be treated by these relatively expensive facilities is quite limited.

Heavy-ion beams have a high linear energy transfer (LET), that is, a higher amount of energy per particle transferred per unit distance. This increased number of ionization events delivered in a shorter distance interval yields an enhanced probability for double-strand DNA breaks among other effects within a tumor cell; this is related to the biological damage delivered per unit dose by calculated comparison to an equivalent photon dose, and is termed the relative biological effectiveness (RBE) [14].

Moreover, densely ionizing radiation may have biological advantages, due to different cell death pathways and release of cytokine mediators of inflammation. Mitotic catastrophe (a pathway preceding cell death that happens in mitosis or as a consequence of aberrant mitotic progression) is the primary mode of radiation-induced cell death in solid cancers. High-linear-energy-transfer (LET) radiation such as with alpha particles has been shown to result in enhanced chromosome rearrangements and reproductive death [15], due to both the complexity and the absolute number of DNA damage clusters [16,17].

### 3.2. Alpha Particle-Based Radiotherapy

High-LET alpha particles are good candidates for killing tumor cells. Alpha-particle-emitting atoms are used in targeted alpha (α) particle therapy (TAT), taking advantage of the short and highly ionizing path of α-particle emissions. Common α-emitters, including ^225^Ac, ^213^Bi, ^224^Ra, ^212^Pb, ^227^Th, ^223^Ra, ^211^At, and ^149^Tb, are in use [18]. TAT is part of the broader field of radiopharmaceutical therapy (RPT), which has expanded in recent years [19].

Nevertheless, the short penetration of alpha particles in tissues (50–90 microns) limits their use for the treatment of solid tumors.

We developed an efficient and safe intratumoral tumor abolition treatment for solid tumors, utilizing alpha-emitting atoms shed from an Ra-224 source inside the tumor. The treatment was termed “diffusing alpha-emitters radiation therapy (DaRT)”. The technology stems from the intratumoral insertion of Radium-224-coated seeds (^224^Ra, 3.66 days half-life), releasing alpha-emitting atoms that disperse in the tumor and deposit highly destructive alpha particles [20]. The short-distance efficiency of alpha particles inside tissues becomes an advantage due to the local release of energy focused on cancerous cells, sparing surrounding healthy tissues. Basic and translational research revealed that the DaRT modality delayed tumor development, extended survival, and reduced metastatic burden in mice bearing various mouse- and human-derived tumors [21,22]. More importantly, as a monotherapy, DaRT has been shown to induce a systemic anti-tumor immune response following tumor abolition. Recently, clinical trials of patients with squamous cell carcinoma (SCC) of the head, neck, and skin yielded a complete response of over 70% of the tumors [23,24].

Diffusing alpha-emitters radiation therapy (DaRT) is the only known method for treating solid tumors with highly destructive alpha radiation.

### 3.3. Particle and Photon Radiation and Hypoxia

The biological effects of photon radiation are heavily dependent on the presence of oxygen, and this may also affect the stimulation of anti-tumor immune response. In fact, the main mechanism of how low-linear-energy-transfer (LET) radiation induces damage is through the formation of radical oxygen species [25]. One of the leading reasons for radiotherapy failure is tumor hypoxia [26].

The oxygen enhancement ratio (OER) or oxygen enhancement effect in radiobiology refers to the enhancement of therapeutic or detrimental effect of ionizing radiation due to the presence of oxygen. The OER is traditionally defined as the ratio of radiation doses during lack of oxygen compared to no lack of oxygen for the same biological effect. The maximum OER depends mainly on the ionizing density or LET of the radiation. Radiation with higher LET and higher relative biological effectiveness (RBE) are less dependent on oxygen in mammalian cell tissues. High LET radiation, such as with alpha particles, has been shown to have OER values of almost 1, which indicates that oxygen has almost no effect on cellular sensitivity to radiation.

Studies have illustrated that heavy ions overcome tumor radioresistance caused by Bcl-2 overexpression, p53 mutations, and intratumor hypoxia, and possess antiangiogenic and antimetastatic potential. [27]. Following photon irradiation, survival and viability of normoxic cells were significantly lower than those of hypoxic cells at all doses analyzed. In contrast, cell death induced by alpha emitter Bi-213 anti-EGFR-MAb turned out to be independent of cellular oxygenation [28]. Furthermore, Studies showed that high-LET α-particle-emitter Ra-223 is more suitable for the treatment of hypoxic tumor cells than irradiation with an Auger electron/γ- or the low-LET beta emitter Re-188 [29].

Carbon ions, owing to the direct DNA damage mechanism they employ, are also relatively cell-cycle- and oxygenation-independent, and can be used to treat hypoxic and radioresistant disease [30]. Direct comparison of photon, proton, and carbon ion radiation effects under normoxic and hypoxic conditions was performed by Huang and collaborators. Four human tumor cell lines were irradiated with 4 Gray (physical dose), and all types of radiation could significantly inhibit the colony formation of tumor cells under normoxia. However, the efficacy of photon and proton radiation was impaired under hypoxia. Carbon ion radiation could still inhibit colony formation [31].

Although the general claim is that high-LET damage is less sensitive to oxygen levels, it was reported that DNA-repair-deficient cells were more sensitive to high-LET radiation under hypoxic conditions than in wild-type controls. Findings suggest that the repair of high-LET radiation-induced damage under hypoxic conditions requires not only the HR repair pathway, but also poly (ADP-ribose) polymerase (PARP). This study suggests that DNA repair inhibition may be a potential strategy for increasing the effectiveness of carbon ion radiotherapy when targeting the hypoxic regions of a tumor [32].

In contrast to the above-mentioned studies, it was claimed that oxygen has no direct influence in radiation-induced DNA damages by different radiation qualities and hypoxia does not limit DNA damage induced by Ra-223, Re-188, or Tc-99m. Dose-dependent radiation effects were comparable for alpha-emitters and both high- and low-energy electron emitters [33].

## 4. Anti-Tumor Immunity Can be Triggered by Radiation Therapy-Mediated Tumor Abolition

RT is widely used with curative or palliative intent in the clinical management of multiple cancers. Although mainly aimed at direct tumor cell killing, mounting evidence suggests that radiation can alter the tumor to become an immunostimulatory milieu. Early reports described the elimination of nonirradiated lesions following photon irradiation of other tumor lesions. The phenomenon was termed the “abscopal effect” [4,5], an effect which was later attributed to the induction of anti-tumor immunity [34]. Abscopal effects due to irradiation alone remain rare phenomena in the clinics and involve a balance of radiation’s immunogenic and immunosuppressive effects. Clinically, if radiation treatments can be optimized to promote anti-tumor immunity, this could increase the odds of achieving local cancer control and combat growth of micrometastases.

A considerable number of reports addressed this issue, and experimental data could indicate that the photon radiation-induced tissue damage triggers production of generic “danger” signals that mobilize the innate and adaptive immune system. The danger microenvironment engenders a DC-mediated antigen-specific immune response [35,36,37]. Several review articles gathered information about the impact of RT on tumor immunity, including tumor-associated antigens, antigen-presenting cells, effector mechanisms, and the tumor microenvironment [38,39]. The interactions between radiation and the immune response are complicated, and in order to optimize them it will be required to assess the immune response to radiotherapy at the patient level and find approaches that will predict the interaction of immunotherapy with radiotherapy. This may enable to develop radiotherapy regimens more suitable for combination with immunotherapy [40].

The interrelationship between radiation and the immune response can work both ways. In an interesting literature survey of preclinical and clinical studies, Vanneste and co-workers analyzed the radiation enhancement factor effects of immunotherapy on the local tumor in comparison with other traditional radiation sensitizers. Their results imply that for the same RT dose, a higher local control was achieved with a combination of immunotherapy and RT in preclinical settings. Thus, they suggest the use of combined RT and immunotherapy to improve local tumor control in clinical settings without exacerbation of toxicities [41].

## 5. Activation of Anti-Tumor Immunity by PRT of Tumors

Treatment of cancer using particle radiation raises several important questions:Can alpha particles-, protons-, and heavy ions-mediated destruction of tumors trigger anti-tumor immunityIs particle-based radiation more efficient than photon radiation in this respectCan the radiation-induced anti-tumor immunity be further augmented by manipulation of the immune response

To answer these questions, studies were performed using particle radiation and measuring anti-tumor immunity. Anti-tumor immunity in experimental systems can be best determined in vivo by the induction of resistance to a tumor cell challenge following primary tumor destruction, and appearance of lymphocytes that can kill tumor cells specifically. In vitro, radiation-dependent effects on immune function can be observed through the manifestation of immune-related changes in tumor cells. In humans, the ultimate sign of radiation-dependent immune activation is the abscopal effect.

Enforcement of anti-tumor immunity following alpha-particle-mediated tumor destruction was reported in several studies with different radioactive sources.

Analysis of immune-response-dependent anti-tumor activity following intratumoral alpha particle treatments revealed that the Ra-224-based radiotherapy, DaRT, offers a technique to eliminate local and distant malignant cells, regardless of their replication status, by stimulating specific anti-tumor immunity through the supply of tumor antigens from the destroyed tumor [42]. In a series of experiments, mice bearing weakly immunogenic DA3 adenocarcinoma or highly immunogenic CT26 colon carcinoma were treated by Ra-224-loaded wires (DaRT seeds). In both tumor types, tumor growth was significantly retarded in the alpha-radiation-treated mice and the animals developed resistance to a tumor challenge. In the highly metastatic DA3 model, the treatment reduced the prevalence of lung metastases from 93% in the control mice to 56% in the DaRT group [43].

Alpha particle treatments could enhance the probability of an immune response, which can lead to abscopal effects. In a patient with skin SCC treated with intratumoral Ra-224-loaded seeds, lesion shrinkage was evident after 28 days and complete remission of the treated lesion was observed after 76 days. Two other nontreated distant lesions also disappeared, which could be associated with an immune-mediated response. One year after the treatment, a complete remission of the treated lesion was observed as well as spontaneous regression of untreated distant ones [23].

Using bismuth-213 irradiation of murine adenocarcinoma MC-38, it was shown that a protective anti-tumor response was induced that is mediated by tumor-specific T cells. Thus, α irradiation can stimulate adaptive immunity, elicit an efficient anti-tumor protection, and therefore is an immunogenic cell death inducer [44]. Another demonstration of alpha-radiation-based tumor abolition and immunostimulation was reported by Urbanska and colleague [45]. Nanoparticles engineered to target the melanocortin-1 receptor expressed on melanoma (B16 melanoma) were loaded with the alpha particle emitter, Actinium-225. Treatment of B16-melanoma-bearing mice resulted in changes of fractions of naive and activated CD8 T cells, Th1 and regulatory T cells, immature dendritic cells, monocytes, MΦ and M1 macrophages, and activated natural killer cells, in the tumor microenvironment. The treatment also upregulated the inflammatory cytokine genome and adaptive immune pathways [45].

Proton and carbon ion radiation were also reported to stimulate the immune response after treatment of tumors. Most of these studies report on the increase in immune-response-related components on tumor cells in vitro rather than direct stimulation of specific anti-tumor immunity in vivo.

The expression of HLA-, ICAM-1-, calreticulin-, and MHC-class 1-associated TAAs, which figure importantly in T cell recognition of target cells, was analyzed following proton radiation. Proton radiation of prostate, breast, lung, and chordoma cancer cells upregulated the expression of these elements of immunogenic modulation. Moreover, the degree of upregulation of these molecules was similar to that observed after equivalent exposure to photon radiation [46]. In a similar study, the investigators compared the expression of calreticulin (ecto-CRT) in multiple human carcinoma cell lines following irradiation by proton and carbon ion in comparison to photon radiation. Calreticulin is an important indicator of immune cell death (ICD). All the three types of radiation increased the ecto-CRT exposure, with proton and photon radiation equally effective, while carbon ion revealed a different effectiveness in comparison to photon and proton [47]. Durante and Formenti [48] argue that particle radiation can be more effective than X-rays when used in combination with immunotherapy. Protons and heavy ions have physical advantages compared with X-rays, and lead to a reduced damage to blood lymphocytes that are required for an effective immune response.

Another example for the complicated interrelationship between radiation and immune response components and its effect on tumor development was disclosed in an interesting study by Beheshti and coworkers [49]. They found that murine Lewis lung carcinoma (LLC)-derived tumors develop faster in syngeneic adolescent (68 day) compared with old (736 day) C57BL/6 mice. These differences were further intensified by whole body proton irradiation, with increased inhibition in tumors grown in old mice. Through network analysis, two key cytokines, TGFb1 and TGFb2, were revealed to contribute to the slower tumor advancement observed in the proton-irradiated old mice compared with that in the nonirradiated old mice [49].

In a clinical trial, Brenneman and colleagues [50] presented data about an abscopal effect in inoperable metastatic retroperitoneal sarcoma (RPS) treated with proton radiation. A patient with inoperable, metastatic, unclassified round-cell RPS was treated with palliative proton radiotherapy only to the primary tumor. Following completion of radiotherapy, the patient demonstrated complete regression of all un-irradiated metastases and near complete response of the primary lesion without additional therapy.

## 6. Potentiation of Particle-Radiation-Mediated Anti-Tumor Immunity by Immunomanipulation

In order to maximize cancer elimination and the prevention of tumor escape mechanisms, combinations of particle radiation and immune modulating agents, capable of potentiating the immune response, were tested in preclinical and clinical settings.

These studies include the use of the following:Agents that stimulate immune response components. These include microbial or chemical immunoadjuvants, tumor vaccines, and cytokines. Such immunostimulators can promote the activity of dendritic cells and/or T lymphocytes.Agents that inhibit cells and molecules that suppress anti-tumor immune responses. These include agents that inhibit the function or deplete immune suppressor cells such as myeloid-derived suppressor cells (MDSC) or regulatory T cells (Tregs), or inhibitors of the suppressive function of immunological checkpoint molecules (CTLA-4, PD-1, and PD L1).Adoptive transfer of anti-tumor T lymphocytes or antibodies.

### 6.1. Agents Stimulating Immune Response Components

#### 6.1.1. Immunoadjuvants

The Toll-like receptors family is mainly expressed on immune cells, where it senses pathogen-associated molecular patterns and initiates innate immune response. Toll-like receptor (TLR) agonists (ligands) demonstrate therapeutic promise as immunological adjuvants for anticancer immunotherapy. Ligation of Toll-like receptors results in the induction of strong immune responses that may be directed against tumor-associated antigens. Today, 13 distinct TLRs are known to be expressed in mammals (10 in humans), and proteins of the TLR family have been identified in evolutionarily distant organisms including fish and plants.

TLR agonists were included in the National Cancer Institute list of immunotherapeutic agents with the highest potential to cure cancer. To date, three TLR agonists have been approved by U.S. regulatory agencies for use in cancer patients. Additionally, the potential of hitherto experimental TLR ligands to mediate clinically useful immunostimulatory effects has been extensively investigated over the past few years. A summary of recent preclinical and clinical advances in the development of TLR agonists for cancer therapy was published [51]. The effects of TLR stimulation in cancer, expression of various TLRs in different types of tumors, and the role of TLRs in anticancer immunity and tumor rejection were also discussed in a recent review [52].

One of the TLR agonists approved for treatment of cancer is Imiquimod (TLR7 agonist) (a small non-nucleoside imidazoquinoline originally known as S-26308 or R-837). Similar to other imidazoquinolines (e.g., S-27609), imiquimod turned out to act in vivo as a potent inducer of immunostimulatory cytokines, including IFNα, TNFα, and interleukin (IL)-1β and IL-6, and to exert consistent anti-tumor effects.

Unmethylated CpG-containing oligodeoxynucleotides are strong TLR agonists (TLR9) and activators of anti-tumor immunity and of dendritic cell function. CpG was used in many studies in combination with almost all abolition modalities and was found to significantly boost the anti-tumor immune response triggered by the destruction of the tumor by abolition [6]. The intracellular signaling pathways that link TLR ligation with immune activation and where and how TLRs recognize their targets were addressed in the following article [53].

TLR3 recognizes dsRNA or its synthetic ligand polyinosinic:polycytidylic acid [poly (I:C)] and is responsible primarily for the defense against viral infections. The TLR3 agonist poly (I:C) is a powerful immune adjuvant as a result of its agonist activities on TLR-3, MDA5, and RIG-I. Poly (I:C) was developed to mimic pathogen infection and boost immune system activation to promote anticancer therapy. Although TLRs were first identified in immune system cells, recent studies show they can also be expressed in tumor cells.

In preclinical and clinical studies, poly (I:C) and its derivative poly-ICLC were used as cancer vaccine adjuvants and were found to enhance anti-tumor immune responses, and contributed to tumor elimination in animal tumor models and patients [54]. Modified TLR3 agonists (Ampligen^®^, Hiltonol^®^, poly ICLC) are already being used in clinical studies for cancer therapy as single agents or in combination with other drugs. TLR3′s agonists can induce apoptosis and activate the immune system at the same time, making TLR3 ligands an attractive therapeutic option for treatment of cancer [55,56].

Poly (I:C) complexed with polyethylenimine (BO-112) was reported to cause tumor cell apoptosis. Intratumoral treatment with BO-112 of subcutaneous mouse tumors led to remarkable local disease control dependent on type-1 interferon and gamma-interferon [57], and was given to cancer patients in combination with checkpoint inhibitors with promising effects [58]

#### 6.1.2. Agents that Inhibit Immunosuppressive Cells: MDSC and/or Tregs

Host immune cells with a suppressive phenotype represent a significant hurdle to successful immunotherapy of metastatic cancer. Among the suppressor cells, Tregs and MDSC are significantly increased in hosts with advanced malignancies.

Tregs, in most cancers, play a central role in contributing to the progression of the disease. Thus, suppression mechanisms mediated by Tregs are thought to contribute significantly to the failure of current therapies that rely on induction or potentiating of anti-tumor responses. Depletion of Tregs by anti-CD25, anti-FoxP3, or cyclophosphamide may serve to boost anti-tumor immunity [59]. MDSCs are a heterogeneous population of immature myeloid cells that are increased in many cancer types. MDSCs play a central role in suppression of the host immune system through mechanisms such as arginase-1, release of immune-suppressive factors such as reactive oxygen species (ROS), nitric oxide (NO), and cytokines. Blockade of MDSC recruitment by blocking chemokine receptors, differentiation of MDSC to macrophages, and blocking MDSC function were found to be essential for an effective anti-tumor immunotherapy [60].

#### 6.1.3. Inhibitors of Immune Suppression Pathways: Checkpoint Blockade

In recent years, cancer immunotherapy gained momentum when the therapeutic benefit of monoclonal antibodies against immune checkpoints (CTLA-4/CD80/CD86 and PD-1/PD L1) was reported. As a follow-up, the beneficial anti-tumor effects of combining checkpoint inhibitors with various abolition modalities were examined. In 2019, the FDA granted approval for PD-1 inhibition as first-line treatment for patients with metastatic or unresectable, recurrent head and neck squamous cell carcinoma (HNSCC), approving pembrolizumab in combination with platinum and fluorouracil for all patients with HNSCC and pembrolizumab as a single agent for patients with HNSCC whose tumors express a PD-L1. These approvals marked the first new therapies for these patients since 2006, as well as the first immunotherapeutic approvals in this disease [61]. Inhibitors of PD-1/PD-L1 include peptides, small-molecule chemical compounds, and antibodies. Several approved antibodies targeting PD-1 or PD-L1 have been patented with good curative effect in various cancer types in clinical practices. While the current antibody therapy is facing a development bottleneck, some companies have tried to develop PD-L1 companion tests to select patients with better diagnosis potential [62].

Given the inferior response rate of immune checkpoint inhibitors (ICI) therapies, researchers performed extensive work and demonstrated that ICI therapies were influenced by a combination of predictive biomarkers related to genomics, immune checkpoints expression, some characteristics in microenvironment, and gut microbiome [63].

### 6.2. Particle Radiation Therapy in Combination with Immunostimulants Can Achieve a Higher Level of Tumor Control of Primary Lesions and Metastases

In view of the activation of specific anti-tumor immunity following tumor destruction by the alpha-radiation-based DaRT, a series of experiments were conducted to examine how it is feasible to enforce this effect by manipulating the immune response. Combining intratumoral alpha radiation with the TLR agonist, CpG, resulted in a better control of the primary tumor and elimination of lung metastases in mice bearing the weakly immunogenic DA3 adenocarcinoma [42]. In successive studies, the efforts to fortify the potency of the anti-tumor effect, triggered by tumor abolition with Ra-224-loaded seeds, were carried out with two approaches: (1) neutralization of immunosuppressive cells such as regulatory T cells (Tregs) and myeloid-derived suppressor cells (MDSCs) and (2) boosting the immune response by immunoadjuvants. Ra-224-loaded seeds were inserted into DA3 mammary adenocarcinoma tumors, and the mice were also treated with the MDSC inhibitor (sildenafil), or Treg inhibitor (cyclophosphamide at low dose), or the TLR-9 agonist, CpG, or a combination of these immunomodulators. A combination of all four therapies led to a complete rejection of primary tumors and to the elimination of lung metastases. The treatment with DaRT and Treg or MDSC inhibitors (without CpG) also resulted in a significant reduction in tumor size, reduced the lung metastatic burden, and extended survival compared with the corresponding controls [64].

A similar approach was taken in a study in which immunomodulatory strategies to boost the anti-tumor immune response induced by DaRT were investigated in the colon cancer CT26 mouse model. DaRT used in combination with the TLR9 agonist CpG, TLR3 agonist, poly I:C, or with the TLR1/2 agonist XS15, retarded tumor growth and increased tumor-rejection rates, compared with DaRT alone. Alpha radiation with CpG or XS15 cured 41% and 20% of the mice, respectively. When DaRT was applied in a combination with CpG, the Treg inhibitor cyclophosphamide, and the MDSC inhibitor sildenafil, the cure rate increased for 41% to 51% of the animals. Cured animals rejected a challenge of CT26 cells but not DA3 (breast cancer) cells, and by passive transfer experiments it was shown that cured mice harbor specific anti-CT26 lymphocytes. [65]. The above-mentioned studies were expanded to additional tumor cell models and immunostimulators. Triple-negative breast cancer (4T1)-, pancreatic (Panc02)-, and squamous cell carcinoma (SQ2)-derived tumors were exposed to Ra-224-loaded DaRT seeds and immunostimulation. Intratumoral delivery of poly (I:C)-polyethylenimine (poly (I:C)-PEI) was used to activate RIG-1-like receptors (RLRs), and poly (I:C) without PEI was used to activate TLR. Poly (I:C), both with or without PEI, prior to DaRT retarded the growth of the tumors and elicited specific anti-tumor activity. Treatments with a T-regulatory cell inhibitor or the epigenetic drug, decitabine, intensified the anti-tumor manifestations of the combination of DaRT and poly (I:C)-PEI and extended survival rates due to lung metastasis clearance [66].

In a recent study, we examined tumor destruction and activation of systemic anti-tumor immunity in mice bearing murine squamous cell carcinoma (SQ2) solid tumors by Ra-224-loaded seeds in combination with either poly (I:C)-PEIor anti PD-1 or both. Tumor development was recorded, and anti-tumor immunity was assessed. Subcutaneous Ra-224-loaded seeds (DaRT) and anti PD-1 effectively retarded tumor progression compared with DaRT alone, and the strongest effect was achieved by combination of DaRT and Poly (I:C), and anti PD-1.

The anti-tumor effects of alpha-radiation and immunomanipulation were also validated in an experimental system of mice with multiple myeloma murine model that express the tumor antigen CD138 and ovalbumin (OVA). The animals were treated with the alpha emitter, bismuth-213, coupled to anti-CD138 antibody, followed by an adoptive transfer of OVA-specific CD8+ T cells (OT-I CD8+ T cells). A significant tumor growth control and an improved survival in the animals treated with the combined treatment were observed [67].

In an important study, carbon ion and photon radiation were compared as to their capabilities to stimulate anti-tumor immunity alone and in combination with checkpoint inhibitors. Mice with advanced osteosarcoma (LM8) carried two tumor lesions, and one of them was irradiated with either carbon ions or X-rays in combination with two immune checkpoint inhibitors (CPI) (anti-PD-1 and anti-CTLA-4). The combined protocol of carbon ions and the immune checkpoint inhibitors administered sequentially was the most effective in retarding the growth of the nonirradiated tumor (abscopal tumor). The combination of immunotherapy with both radiation types essentially suppressed metastasis, with carbon ions being more efficient. Carbon ions treatment alone also reduced the number of lung metastases more efficiently than X-rays. Examination of the abscopal tumors in animals treated with radiation and CPI combination revealed an increased infiltration of CD8+ cells [68].

## 7. Summary

The studies summarized in this review show clearly that particle-radiation-mediated abolition of solid tumors can promote specific anti-tumor immunity in experimental animals and the manifestation of abscopal effects in cancer patients. Furthermore, such immune responses can be boosted by immunoadjuvants, by inhibition of immune suppressor cells and by checkpoint inhibitors that facilitate the functionality of anti-tumor immune cells. Such activities of the immune response act to remove residual tumor cells in the tumor sites and remote metastatic loci.

Whether high-LET particle radiation is better than low-LET radiation in turning the tumor into an immunogen is still an open issue. Yet, the findings that particle radiation can exert its effects under hypoxic conditions is an advantage also from the perspective of anti-tumor immunity facilitation. Another point to consider, although it needs to be substantiated from the immunological point of view, is that particle radiation might cause less damage to surrounding tissues and blood vessels that bring immune cells to the tumor site.

Thus, intratumoral alpha radiation, proton radiation, and carbon ions should be highly considered for treatment of metastatic cancer in combination with immunomodulatory agents.

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
