# Peer review of "The Potentiation of Anti-Tumor Immunity by Tumor Abolition with Alpha Particles, Protons, or Carbon Ion Radiation and Its Enforcement by Combination with Immunoadjuvants or Inhibitors of Immune Suppressor Cells and Checkpoint Molecules"

_cells, 2021, doi:10.3390/cells10020228_

Round 1

Reviewer 1 Report

This is an interesting and up-to-date review about a topic of ever increasing interest. I just  suggest a careful review of the text. For example, the sentence over lines 22-24 needs to be addresssed. 

Reviewer 2 Report

The review provides an interesting overview and motivation to further investigate the combined use of charged particle therapy and immunotherapy as a novel cancer treatment strategy. A few comments and suggestions to improve the manuscript can be found below.

As a first general comment, the manuscript is scientifically sound, but language and sentences should be improved in certain sections of the manuscript. Please pay attention to consistency in terminology and abbreviations.

L16-25: I understand that there is a word limit on the abstract and the last paragraph of the abstract (L26-36) covers the topic well and is well-written in my opinion, but the first part is a bit vague. The main (dosimetric) advantage of charged particle therapy is missing and I would make it clear from the start which form of alpha particle therapy (DaRT) the authors are discussing in the review. Adding this information will make the abstract more sound and attractive.

L20: Gamma radiation (cobalt-60 or cesium-137) is rather outdated and I think the authors refer to photon-based radiotherapy. While this formulation is not wrong, I would recommend to use the same wording throughout the manuscript and change this to photons (gamma-rays and X-rays).

L31: The section between brackets ‘(immunomanipulation)’ can be deleted.

L42-44: Reference is missing.

L45: What do the authors mean with “biological therapy”? Please elaborate on this and/or clarify this in the manuscript.

L48: Does the primary purpose of systemic therapy not include local tumor control as well, next to the focus on metastases?

L56: Replace 'making the tumor its own vaccine' by 'where the cancer vaccine is generated in vivo without the need to identify and isolate tumor associated antigens' and add a reference to this sentence to guide the reader to a relevant paper on in situ vaccination (e.g. Hammerick et al. – Molecular Oncology 2015)

L56: Wasn’t the abscopal effect observed with EBRT before in situ vaccination became available? I think the authors try to say that the systemic T cell response after in situ vaccination will be similar to the abscopal effect observed with radiation therapy, so please reformulate this sentence to bring the message in the correct way.

L80: To be consistent with the abstract, add gamma-rays next to the X-rays between the brackets. Or choose a different description, as long as it is consistent.

L81: While I do understand why the authors add alpha particles to the list, it’s a bit confusing to see them in the list with particles used in EBRT. While experimental studies are ongoing to re-introduce helium beams in the clinic, I don’t think the authors are referring to EBRT here? Could you add and describe the alpha brachytherapy (DaRT) application? For readers with a nuclear medicine background, this might be confusing and it would be good to make a clear distinction with upcoming alpha particle emitting isotopes (e.g. Actinium-225, Astatin-211, Bismuth-212…) which are considered for targeted radionuclide therapy (or targeted alpha therapy).

L82: Use conventional EBRT in these sentences, since proton, neutron and carbon ion therapy are also forms of EBRT. Also in L84: The effectiveness of conventional EBRT is limited for hypoxic tumors.

L105-106: Be consistent in terminology for external photon therapy, EBRT, RT…

L113: Similar to previous comment, be consistent in terminology: charged particle beam radiotherapy, PRT…

L98-112: In general, this section is quite long and the readership of this special issue might be less familiar with this description. Therefore, I would recommend to shorten this section and limit the description to the main characteristic that makes charged particle therapy beneficial over conventional photon-based RT. Namely the dosimetric advantage of the Bragg peak which provides the opportunity to spare surrounding normal tissue and decrease the integral total dose. In addition, the lack of an exit dose makes it possible to treat tumors close to critical structures.

L120-125: The higher RBE of charged particles was already mentioned in L96-97, try to bring these sections closer together.

L126-127: Add a reference for this sentence.

L129: Replace ‘context’ by ‘mode’.

L129-132: Are chromosomal rearrangements the most interesting biological end point to look at for tumor control/ablation (given the aneuploidy of many cancer cells)? Wouldn’t it be more relevant to add references to different modes of cell death here…

L143: Replace ‘retarded’ by ‘delayed/inhibited’

L148-149: I think quite some of the alpha-particle emitting radiopharmaceuticals are already in pre-clinical trial as well?

L190: Perhaps delete “mediated tumor ablation” from the title to make it a bit shorter?

L191: The RT abbreviation was previously used for the first time.

L194-195: Reformulate this sentence so it reads better.

L200: Be consistent in the use of photons, gamma-rays or X-rays. Or define it clearly at the beginning of the paper.

L204-208: Very long sentence, please shorten/split it.

L216-217: I would suggest to replace ‘treatment of tumors’ by ‘radiotherapy’ in heading 5.

L226: Please explain ‘the induction of resistance to tumor cells’ in this sentence? After being by challenged by the injection of tumor cells?

L228: Briefly elaborate and explain what ‘immune related changes in tumor cells’ in the in vitro experiments entail?

L303: Suggestion to change heading 6.1 to ‘Agents stimulating immune response components’.

L347: Add some examples of strategies (antibodies, small molecules…) for Treg and MDSC depletion, e.g. anti‐CCR4 mAb and others…

L437: It’s not entirely clear why the authors change between ‘abolition’ and ‘ablation’ in the manuscript (also check the title)?

L449: Section 6.2 doesn’t really bring a motivation to use protons in combination with immunomodulatory agents. Are some studies/references missing in this section or are there no studies available (yet)?

Reviewer 3 Report

Dear authors,

this review provides a comprehensive overview and update, respectivley, on the field and will certainly contribute to increase the attention on the vast potential that particle therapy can offer in combination with different immunotherapies. I only have a few minor comments on the manuscript.

For example, I wonder if there is not too much information given generally on particle therapy. The paragraph from line 113 to 119 could e.g. be removed. On the other hand, the journal is not a classical one in the community and the readers might thus get a certain background. There is a certain nuance of a tendency in favor of DaRT, however, the potential conflict of interests has been mentioned. So I am fine with that.

Generally, I feel some parts could need more references, especially in the light of this journal not being specific to the community. While some facts might be trivial for the community of radiobiologist, it could be interesting for readers beyond the field to have such informatin (e.g. the OER, l. 156 etc.).

L. 58: Along with the tumor antigens, the adjuvanticity of radiation plays a major role. That should be mentioned in my opinion.

L. 84: I would state "...mainly due to hypoxia...".

L. 100: The authors describe the active scanning method of CPT, which is indeed increasingly used, up to my knowledge, but is not restricted to it. Passive shielding does still play a role in a few centers, even though it uised less and less.

L. 104: I disagree with the phrasing. It should be pointed out that healthy tissue is actually even better spared with CPT than with photons. I find the sentences somewhat misleading.

L. 120: The entire paragraph is literally similar to the the references. I just want to point at that in order to avoid any conflicts.

L. 164: It states "recent studies", however, the reference is from 2009. That should be adapted.

L. 194: Please point out that abscopal effects due to irradiation alone remain rare phenomenons in the clinics.

L. 209: The effects of the immune response on the local/primary tumor have been shown earlier (e.g. Lee et al. 2009, doi: 10.1182/blood-2009-02-206870)

L. 219: The first question raised should be ammended to state that radiation types need to destroy the tumor and at the same time trigger anti-tumor immunity. I find that important since the balance between TCP and an effective immune response is still to be defined and there is discussion on whether it should be a single high dose or e.g. a hypofractionation.

L. 226: I assume that "induction of resistance" refers to a re-challenge with tumor cells? Please specify.

L. 229: That goes as well for in vivo experiments.

L. 263: TAAs are actually tumor-associated antigens and are not expressed in the surface as such, as the other molecules mentioned. They are, of course, presented on the surface, but I find the sentence misleading.

L. 276: I failed to find the references for the increased release of cytokines also in reference number 40. They should be mentioned or the sentence should be adapted.

L. 277: Although representing very interesting data, that paragraph seems somewhat out of context, or, from another point of view, would need more explanation on why that is important here.

L. 446: I certainly agree on that statement and would like to add that also the immune cells (blood cells) themselves are better spared than with photons. Reference 40 should deal with that aspect if I am not mistaken.
